# Applying Artificial Intelligence to Wearable Sensor Data to Diagnose and Predict Cardiovascular Disease: A Review

**DOI:** 10.3390/s22208002

**Published:** 2022-10-20

**Authors:** Jian-Dong Huang, Jinling Wang, Elaine Ramsey, Gerard Leavey, Timothy J. A. Chico, Joan Condell

**Affiliations:** 1School of Computing, Engineering and Intelligent Systems, Ulster University at Magee, Londonderry BT48 7JL, UK; 2Department of Global Business & Enterprise, Ulster University at Magee, Londonderry BT48 7JL, UK; 3School of Psychology, Ulster University at Coleraine, Londonderry BT52 1SA, UK; 4Department of Infection, Immunity and Cardiovascular Disease, The Medical School, The University of Sheffield, Beech Hill Road, Sheffield S10 2RX, UK

**Keywords:** cardiovascular disease, wearable sensor devices, artificial intelligence (AI), machine learning (ML), deep learning (DL)

## Abstract

Cardiovascular disease (CVD) is the world’s leading cause of mortality. There is significant interest in using Artificial Intelligence (AI) to analyse data from novel sensors such as wearables to provide an earlier and more accurate prediction and diagnosis of heart disease. Digital health technologies that fuse AI and sensing devices may help disease prevention and reduce the substantial morbidity and mortality caused by CVD worldwide. In this review, we identify and describe recent developments in the application of digital health for CVD, focusing on AI approaches for CVD detection, diagnosis, and prediction through AI models driven by data collected from wearables. We summarise the literature on the use of wearables and AI in cardiovascular disease diagnosis, followed by a detailed description of the dominant AI approaches applied for modelling and prediction using data acquired from sensors such as wearables. We discuss the AI algorithms and models and clinical applications and find that AI and machine-learning-based approaches are superior to traditional or conventional statistical methods for predicting cardiovascular events. However, further studies evaluating the applicability of such algorithms in the real world are needed. In addition, improvements in wearable device data accuracy and better management of their application are required. Lastly, we discuss the challenges that the introduction of such technologies into routine healthcare may face.

## 1. Introduction

Cardiovascular disease (CVD) is the commonest cause of mortality worldwide [1,2,3,4,5]. Digital healthcare encompasses personalised health and medical information, devices, systems, and platforms, and the integration of comprehensive medical services [6,7]. This has the potential to improve the prevention, diagnosis, treatment, and management of diseases by linking healthcare with information and communication technologies [7,8]. The early identification and prediction of CVD is critical for improving healthcare outcomes, but this remains challenging. The development of a variety of sensors, communication networks, and wearable and portable devices has enabled the acquisition of multiple forms of data that have the potential to improve the diagnosis of CVD [7,8,9,10,11,12]. The manual analysis of often complex time-series data is difficult, requiring the development of algorithms that analyse such data to potentially enhance diagnosis. 

Artificial Intelligence (AI) is a powerful technology that can be used to address challenging tasks in a wide range of real-world domains. Computer systems can be developed to perform advanced tasks such as modelling non-linear and extremely complex systems in the fields of visual and speech recognition, translation between languages, fault detection in industrial manufacture, construction industry innovations that are helpful for challenging urban sustainability [13,14,15,16], medical diagnostics for cost-savings and decision-making, etc. Recently, a chemical engineering problem was solved by predicting levulinic acid synthesis from sugarcane bagasse, which provides a prospective commercialization in market segments by replacing petroleum-based chemicals [13]. 

The application of Artificial Intelligence (AI) in the deployment of wearable sensors is an indispensable part of the revolution of the Internet of Things (IoTs) [14]. Consequently, the rapid development of AI techniques has provided health professionals with a greater capacity to deal with the huge amounts of data collected through wearable devices that have been deployed in monitoring patients’ health conditions. AI improves the ability to explore the relationships between the information acquired from the sensor signals’ output and the health status of individuals through the establishment of various types of diagnostic and predictive models. Understanding AI applications in wearable sensor data manipulation is, therefore, of paramount importance in optimising the diagnosis and prediction of CVD.

This review summarises the literature on the application of AI to wearable sensor data to predict and diagnose CVD. In the following sections, the methodology applied to search the literature is described. Thereafter a detailed depiction of the dominant AI approaches presently applied for modelling and prediction using data acquired from wearables is presented. We describe the types of sensor data used, the AI algorithms applied and the diseases that AI have dealt with. The technical capabilities of wearable devices, the clinical indications and the challenges, limitations and future opportunities of this technology are then discussed. 

We conclude that AI and machine-learning-based approaches are superior to traditional statistical methods for predicting cardiovascular events. However, further studies evaluating the applicability of such approaches in the real world, and improvements in wearable device data accuracy and better management of their application, are necessary for these technologies to fulfil their promise to enhance cardiovascular healthcare.

## 2. Methodology

Through the research, we sought to identify: 

(a)Current scientific advances in wearable device applications in CVD diagnosis and prediction with AI in the digital healthcare domain;(b)AI tools used in diagnostic practice; and(c)Current challenges and future developments of AI in cardiac disease management.

The search considered the PRSMA protocol [17,18,19] and procedures, with details described as follows.

### Search Strategy

Five digital sources were used for the identification of relevant studies. Web of Science and Scopus were selected to retrieve studies, as these are the leading digital literature databases of peer-reviewed research, including extensive scientific and interdisciplinary work. PubMed and Cochrane Library were also searched due to their medicine background. As the newest research outcomes often appear quickly in Google Scholar with a broad range of disciplinary coverage, this was also selected as one of the literature sources. 


**
*(1) Inclusion criteria*
**


(1.1) **The key words group**: the selection of studies aimed to cover four aspects; wearable devices, clinic diagnosis, cardiac diseases, AI techniques such as machine learning and deep learning, and modelling (with a consideration of including prediction related studies). Specifically, with the use of parentheses and Boolean operators to restrict the range of the output, these were a). “*clinic** AND *diagnos** AND (*cardio** OR (*heart disease*) OR (*heart failure*) OR *valve* OR (*atrial fibrillation*) OR *angina* OR (*myocardial infarction*))”, (2). *“smartphone* OR *wearable* OR *sensor* OR *monitoring* OR *remote”,*(3). “(*Artificial Intelligence*) OR (*Machine learning*) OR (*Deep learning*) OR (*Neural Network*)”, and (4). “Model*”. Finally, combination of these fields using AND operators was applied.

(1.2) **Year coverage**: as we sought to investigate most recent developments in the area, and the sharp increase in the number of the relevant publications from 2020 to 2022, the selection criterium covered a period from January 2018 to April 2022. Some exceptionally relevant studies outside this period, which were deemed important to include and were cited in publications identified within the specified date range, were also included.

(1.3) **Language**: the searched literatures were all published in English.


**
*(2) Exclusion criteria*
**


(2.1) Duplicated papers: results extracted from different databases can be duplicated, and any such occurrences were excluded.

(2.2) Papers with content beyond the scope of the planned collection domain: some results of the search were out of the needed range. For example, a paper appeared that met all the keywords and syntax for the search but was found to not be relevant. This type of occasional occurrence was dealt with manually rather than using a NOT operator, so that as wide as possible a range of studies was considered for inclusion.

(2.3) Google Scholar searches do not guarantee full matches with all the keyword combinations in all its outputs. The results usually matched better in front output pages than in later pages. Those studies with less relevance were excluded, i.e., we adopted those that contained the majority of the key words and discarded those that only contained one or a few key words. 


**
*(3) Output of the searches*
**


The above search strategy identified more than 300 papers. Further screening was performed to consider the relevance of the work, journal impact factor and the number of citations of the paper. This resulted in about 135 papers being selected for inclusion in this review.

## 3. Results

### 3.1. Types of Sensor Data

Many different types of data relevant to CVD can be obtained from a wide range of sensors. These include smartphones, wrist-worn wearables or adhesive vests, socks, shirts and accessories such as smartwatches, patches, wristbands, rings and glasses. A mainstay of clinical cardiology is the electrocardiogram (ECG), which collects and records the electrical signals generated during the cardiac cycle via electrodes placed on the limbs and chest. Wearable ECG monitors are now present in many devices and can obtain both single-lead and multi-lead ECGs, either continuously or by being user-activated at specific times. 

Photoplethysmography (PPG) uses optical sensors to measure pulsatility in skin blood vessels, allowing for the measurement of heart rate (HR), blood oxygen saturation (SaO2) and other cardiac parameters. Accelerometry and GPS-based sensors can provide data on activity, including sedentary time, step count, speed, impact force, exercise, etc. Many commercially available devices now provide these types of data simultaneously [8,11].

### 3.2. Artificial Intelligence Tools used for Cardiovascular Disease Diagnosis/Prediction, Especially for ECG

Artificial intelligence (AI) is a technology and computer system capability that can mimic the intelligence of human learning and problem-solving for complex tasks. Both classical machine learning and deep learning are branches of AI. Machine learning (ML) is the process of using mathematical models to allow a computer to automatically learn from data without direct instruction and improve learning based on experience. Machine learning works on algorithms and uses a large amount of data to make decisions and the generated model can produce proper results or give predictions. A neural network (NN) is one way to train a computer to mimic the human brain, which enables the computer system to achieve AI through deep learning (DL). In recent years, there has been a surge in research publications where AI has been used in medicine and healthcare [20,21,22,23,24,25,26,27,28,29,30,31,32,33,34,35,36,37,38,39,40,41,42,43,44,45,46,47,48,49,50,51,52,53,54,55,56,57,58,59,60,61,62,63,64,65,66,67,68,69,70,71,72,73,74,75,76,77,78,79,80,81,82,83,84,85,86,87,88,89,90,91,92,93,94,95,96,97,98].

Existing approaches to machine learning normally consist of three main phases: pre-processing; feature extraction/feature selection, and classification/prediction using artificial intelligence techniques; see Figure 1. In the following subsection, a description of each phase is added for clarification.

*(1)* 
*Preprocessing*


Given raw input signals such as ECG waveforms, a pre-processing procedure is normally required to clean up the data. Subsequently, ECG windows per lead might need to be sampled.

Data cleansing: Data cleansing usually involves removing duplicate records, spelling errors and unreal data. For example, ECG readings in each lead are filtered to remove artefact or noise.

Missing value imputation: In practice, there are many reasons for missing values in the collected signals, such as inconsistent device use, charging time, etc. Missing readings might also be caused when the electrodes are loosely mounted during the acquisition of ECG readings. To obtain an evenly spaced time series, missing data need to be removed or imputed. 

Filtering: This aims to minimise and remove noise using approaches such as electromyogram (EMG) noise, baseline wandering (BLW) and power line interference (PLI).

A first-order low-pass filter and waveform transform can be used to accomplish signal denoising and outlier correction. Noise or motion artefacts that have low- and high-frequency characteristics are discarded using a band-pass filter. The wavelet function in the wavelet transform is used to convolve with the signal and can maintain specific detailed time-frequency components of medical signals such as ECG [99]. In addition, wavelet coefficients can be extracted, and signals can be reconstructed to eliminate the baseline [100]. 

*(2)* 
*Feature extraction:*


The pre-processed data are noise-free, clean and can be used for feature processing. Feature extraction and feature selection transform raw data to features with or without reduced dimensions. Classical machine learning approaches rely on hand-engineered features (see Figure 2, which illustrates types and methods used to extract features). For example, an approach has been proposed for the detection of AF rhythms in ECG recordings [20]. In this case, thirty multi-level features that include 20 morphology features, 4 AF features, 2 RR interval features and 4 features relating to similarity index between beats were extracted. 

The peak point of the R wave is set as a reference point. The intra-beat interval is the interval between the reference points of post-heartbeat and front heartbeat such as the PR interval (see Figure 3, which illustrates an example of ECG beats), which is the interval between the starting point of the P wave and QRS complex of a given heartbeat; the QRS complex combines the R wave, Q wave and S wave, indicating ventricular depolarization; inter-beat interval relates to two consecutive heart beats, the relevant characteristics of the RR interval. 

It has been reported that local statistical information and signal amplitudes are suitable to extract using small-scale convolution filters [101]; for example, the amplitudes of R, S, P and T waves in an ECG signal, where the interval characteristics among waves and morphological features are suitable for extraction using large-scale convolution filters embedded with bigger receptive fields, such as R-R interval, P-R interval and QRS duration. As noise and artifacts can dampen atrial signal, and can even make the signal not exist, atrial-activity-related features are more challenging to extract. In [45], a novel and simple way to extract features was proposed by utilizing tenable quality factor (Q-factor) wavelet transform (TQWT), phase space reconstruction (PSR) and variational mode decomposition (VMD). 

To improve the performance of the trained model, unrelated features in the dataset were discarded. However, it is challenging to reduce the number of features and achieve accurate disease prediction with reduced features. Therefore, feature selection techniques were applied to select the most relevant features to diseases. Different feature selection techniques exist in the literature such as a sequential forward selection (SFS) technique which relies on an extensive selection strategy to estimate the performance of each feature separately [34], feature selection based on the Levy crow search algorithm after missing data were imputed and data were normalized [47]. While in [22], feature selection was based on the importance of 55 features, the decision tree (DT)-based model was created for CAD diagnosis, with the five most important features used for classification. 

One of the advantages of DL is that raw data are the inputs for the deep neural networks. Feature extraction and feature selection can be automatically performed during training, and the manual engineering of features, therefore, becomes less necessary. In deep learning, convolutional neural networks are usually applied to extract time-invariant features automatically, and the important segments of input signals are more emphasised using an attention mechanism, which is most likely to contribute to raising an alarm. Long short-term memory units are applied to capture the temporal information that exists in the signal.

*(3)* 
*Artificial intelligence, machine learning and deep learning*


Wearable devices are rapidly developing in the health fields for telemedicine, patient monitoring, and mobile health (mHealth) systems. The role of these devices has been examined in the remote monitoring and diagnosis of common CVDs [10,11,102], and the opportunity and obstacles of these devices have been explored [103,104]. Specific barriers and knowledge gaps such as HR and activity tracking have been identified for the use of wearables in clinical cardiovascular healthcare [105]. A Heart Health Monitoring Service Platform (HHMSP) has been proposed by combining the Internet of Things (IoT), HR measurements and advanced AI for multi-disease monitoring [106]. The strengths and weaknesses of using new technologies such as cloud models, IoT sensors and AI in assisted-living environments to improve medical services have been identified as helping to reduce the dependence on traditional healthcare systems and services [24].

Many algorithms in the medical and health fields have been presented, which can be categorized as classical machine learning and deep learning methods. Review papers from different angles have been published [107,108,109,110,111]. 

The current state of AI in clinical applications has been described in several reviews [112,113,114]. AI is a potential solution for precision medicine that can be tailored to the needs of the individual patient. However, more efforts should be made to make precision medicine closer to reality by combining electronic health records (EHR), patient-level ambulatory sensor data, and genomics using ML analytics. The impact of AI and recent advanced technologies have been explored in all aspects of arrhythmia care. There are challenges regarding imbalanced ECGs and patient data in heart diseases [115]. The mathematical background related to supervised AI algorithms have been described and selected, and AI ECG cardiac screening algorithms discussed [116]. 

DL has been an exciting innovative research area over the years, and the challenges and opportunities of this area in cardiovascular medicine have been overviewed [111]. A critical review of the strengths and potential limitations of DL approaches is included. End-to-end DL can be used for the analysis of resting ECG signals to detect structural cardiac pathologies, which can be applied to the screening of asymptomatic populations effectively [117].

Various kinds of anomalies using smartphone sensors in healthcare have been detected. The main limitations and advantages of the use of smartphone sensors systems are listed [118]. In relation to risk prediction models in CVD, the biomarkers can be used for early detection of the disease as well as risk predictions [119], ML and AI provide a new landscape of real-time stroke prevention in the digital health field [120].

A review of classical machine learning and deep learning methods is presented below.

Classical machine learning algorithms

Classical machine learning methods have been widely used in the detection and pre-diction of heart diseases by combining feature extraction based on prior physiological knowledge. The methods include supervised ML such as support vector machine (SVM), decision tree (DT), random forest (RF), k-nearest neighbours (KNN), unsupervised ML such as hidden Markov model (HMM), principal component analysis (PCA), ensemble and other rule-based or statistical approaches (see Figure 4).

Table 1 lists the relevant works of the identified studies and details in terms of the authors and databases used, types of diseases and data, algorithms developed, areas of application, and evaluation performance. Metrics such as accuracy (ACC), specificity (SPE), sensitivity (SEN), true positive rate (TPR), true negative rate (TNR), receiver operating characteristic curves (ROC), area under the ROC curves (AUCs) and F1 score, are used to evaluate the performance. A wide range of heart-related diseases have been explored, for example, atrial fibrillation (AF), coronary artery disease (CAD), hypertrophic cardiomyopathy (HCM), congestive heart failure (C-HF), heart failure with reduced ejection fraction (HFrEF), left ventricular systolic dysfunction (LVSD), myocardial infarction (MI), acute ST-elevation MI (STEMI), stable ischemic heart disease (SIHD), sinus rhythm (SR), ventricular arrhythmias (VA), asymptomatic left ventricular dysfunction (ALVD), arterial blood pressure (ABP), mitral regurgitation (MR), aortic stenosis (AS), paroxysmal supraventricular tachycardia (PSVT), heart failure with preserved ejection fraction (HFpEF), and ventricular ectopic beats (VEB).


*(a.1) Supervised learning*


There are more ML techniques that belong to the category of supervised learning. SVM is a supervised method that has been widely used in medical and health data analysis, and is an efficient approach to classify data in a hyperplane. The decision tree has the structure of a tree, and can be learned automatically from data according to a certain parameter. The random forest approach enables higher resolution, since more and smaller regions are formed in the feature space. Diversity among the trees is created by learning on a random sample and random features for splitting at each node. Figure 5 presents an example for each of the mentioned ML techniques.


*(a.2) Unsupervised learning*


In unsupervised learning, there may not be an output for every input; thus, the desired outputs are unknown. The goal is to try to identify hidden patterns in such unlabeled inputs. Unsupervised learning algorithms are usually applied to solve tasks related to dimensionality reduction, clustering, and outlier/anomaly detection. They have been used for detection purposes on health care datasets. HMM is a class of a probabilistic graphical model that is frequently utilised for modelling biological sequences data. It allows for the prediction of a sequence of hidden metrics from a set of observed metrics. In [29], an ML model was presented to classify patients’ self-reported physical health using activity tracker data with stable ischemic heart disease (SIHD). An HMM model was constructed by utilising correlations between successive weeks; the proposed model achieved an AUC of 0.79 for classifying health status over time, and activity trackers can be used to monitor patient outcomes in real time.


*(a.3) Ensemble approaches*


The ensemble approach combines multiple base models to obtain optimal predictive performance. A reliable and high-performance system for heart arrhythmia classification has been developed with physiological meaning and low cost [40], with a novel visual pattern and a combined parametric feature of ECG morphology, a combination of KNN, SVM, and a neural network framework presented for evaluation. A novel approach [39] has been developed for the diagnosis of aortic stenosis (AS, narrowing of a major valve in the heart) using the time-frequency features extracted from cardio-mechanical chest signals (SCG and GCG). The analysis of variance test was used to select the features, and the DT, RF, and ANN methods were used to evaluate different combinations of features. A wearable ECG telemonitoring system [50] was proposed for atrial fibrillation (AF, the commonest heart rhythm disorder) detection, in which a smartphone and cloud computing were used. ECG signals were obtained using a designed wearable ECG patch, with the signals sent to an Android smartphone via Bluetooth. The ECG waveforms were illustrated in a developed Android APP in real time, and every 30s ECG data were transferred to a remote cloud server. The CatBoost-based machine learning method has been proposed for the detection of AF in the cloud server and classification results and the ECG data are pushed back to the clinician’s web browser. Finally, the clinician’s diagnosis for the ECG signals is displayed on the Android APP. In addition, valuable editorial comments [36] have been provided [45], in which AF events can be predicted four hours before the attack of the arrhythmia using PPG data collected from the Huawei Heart Study, XGBoost was used to develop the ML-based model for future AF prediction and provided an opportunity for more effective approaches that could provide the early detection, and treatment and prevention of cardiac diseases.


*(a.4) Other approaches*


Hypertrophic cardiomyopathy (HCM) is a genetic disease causing thickening of the heart muscle, heart failure and life-threatening rhythm disoders. An approach used to detect HCM via PPG signals is the automated multiple-instance ML model via an embedded instance selection model [44]. Features that relate to hemodynamic abnormalities in HCM, such as systolic ejection time and the rate of systolic pressure rises, combined with extracted morphological features are used for detection. In [30], a smart wearable system named Cardiovascular Disease Monitoring (CVDiMo) was developed for cardiovascular risk assessment in the short-term, with emotional dynamics. Six different bio-signals from two different test groups with 30 participants were tested and analysed. In addition, stress levels deduced from the emotional state analysis integrated with the physical activity results achieved a higher performance in risk estimation.

However, in traditional machine learning fields, morphological features, such as the QRS complex, which are extracted from ECG or PPG signals, popular signal processing techniques such as wavelet transform and Fourier transform, are used in most of the classical algorithms for training and validation of algorithms. The features vary significantly under different conditions and among individuals and are not enough for arrhythmia detection and improved accuracy.

b.Deep learning (DL)

In recent years, AI approaches with DL algorithms (see Figure 6) such as deep neural networks (DNNs), convolutional neural networks (CNNs), spike neural networks, recurrent neural networks (RNNs) and its popular extension, long short-term memory (LSTMs) networks, have all been widely used to interpret ECG signals. DL models aim to extract the features from the raw data and perform the classification in a single framework.

Deep learning has also been successfully used in time series processing. DNNs can build computational models that include multiple hidden layers; data are learned with various kinds of abstraction. DNNs are more powerful with deeper networks, which can be trained more effectively with much faster hardware, and their performance is greatly improved as a large number of context-dependent output units are used. DNNs are capable of approximating nonlinear functions and solving tasks by using classical linear methods. However, it is well known that the DNNs are computationally intense and consume large amounts of power; in addition, parameters such as the number of neurons in each layer, the number of hidden layers, the optimizer, the activation function, the ways to prevent overfitting, etc., need to be optimised in the DNNs approach. This imposes certain challenges for efficient learning in real-time stream processing with embedded multiprocessors with low power. In recent years, many solutions in AI and DL fields have been proposed.

Table 2 lists the performance comparison of relevant works that involve AI and deep learning systems for the detection and diagnosis of cardiovascular disease, which are listed in the order of ANNs, RNNs, LSTMs, SNNs, CNNs and hybrid approaches.

In the following subsections, we briefly review each of these learning approaches.


*(b.1) Recurrent neural networks and long short-term memory neural networks*


RNNs have the advantages of dealing with sequence dependence between consecutive windows and can perform better with short-term forecasting when modelling time series data. However, it is not easy for RNNs to effectively train long sequences as it causes more difficulties in transferring information from the previous steps to later ones, and may encounter vanishing and exploding gradients when procedures such as backpropagation-through-time (BPTT) are trained iteratively on long sequences [121]. LSTM networks using LSTM cells can better find long-term dependencies. RNNs LSTM architecture is suitable for exploiting ECG signals that have time-series-based sequential data structures [122]. The ECG waveform can be classified from one timepoint to the next for each heartbeat using an LSTM network with ECG signal delineation. However, due to the lack of feature extraction in LSTM, the obtained results are less optimal.

Over a long enough monitoring period, the atrial fibrillation burden (AFB) is related to the ratio of time in AF. It has been suggested that using the AFB can add prognostic value compared to a binary diagnosis. A deep RNN learning model, ArNet [57] outperformed a gradient boosting (XGB) model and could feasibly estimate AFB from a one-day beat-to-beat interval time series. Robust remote diagnosis and phenotyping of AF are possible. A model [59] that employed a novel structure consisting of wavelet transform and multiple LSTM-RNNs networks has been proposed for continuous cardiac monitoring on wearable devices. Experimental evaluation shows a superior performance compared to existing works and the proposed model is lightweight compared to compute-intensive deep learning-based approaches. A Deep Belief Network (DBN)-LSTM approach [61] for predicting cardiovascular events in advance of a few weeks or months using 5-min ECG recording has also been proposed. DBN can improve training speed and avoid being stuck at local minima. As a feature selection and representation method, DBN can extract high-level features and has the power to discriminate among different classes using unlabeled data. The evaluation results demonstrate that the DBN-LSTM approach can identify patients at high risk of developing heart diseases in the future, by combining heartrate variability measurements analysis.


*(b.2) Spiking Neural Networks*


SNNs represent the third generation of neural networks, while the building blocks in artificial neural networks (ANNs) are based on an abstraction of real biological neurons, lacking a real basis in the biological function of neurons. SNNs mimic the human brain, and, as such, information is transferred using spike signals as well as the timing of the spikes throughout a parallel network. Every neuron consumes energy only when it sends or receives spikes. A novel SNN for real-time ECG cardiac monitoring systems has been proposed for ultra-low-power wearable devices [66]. An unsupervised Spike timing-dependent plasticity (STDP)-based procedure layer with support from two other layers, i.e., the Gaussian and the inhibitory layers, is used to automatically extract features, and the supervised reward-modulated STDP (R-STDP) layer is used to classify the extracted features. Inhibitory neurons in the network prevent the same or similar patterns from having the attention of all neurons in the STDP layer. The learning rules are adjusted to adapt to the ECG classification domain. SNNs provide an opportunity for ultra-low power operation and are suitable for cardiovascular monitoring in real-time in an embedded device.

While most existing works employ complex pre-processing to improve accuracy such as heartbeat R–R intervals [123], higher-order statistics (HOS) [124], and wavelets [125], the raw signals are used to further reduce the energy cost while a comparable performance can be achieved.


*(b.3) Convolutional neural networks*


Convolutional neural networks (CNNs) can extract local features that are used to model the correlation between the spatial and temporal signal and reduce spectral variance; thus, they have strong feature extraction abilities, where features are learned by models trained through backpropagation. Due to the high discriminatory power of CNNs, more tasks employ CNNs as their dominant choice. The experiments reported in [63,93,95] have had successful ECG signal classification using CNNs. However, controlled hospital settings are required in the existing approaches. In [87], a multi-task deep learning method (DeepBeat) was developed for AF detection in real time and signal quality assessment in wearable PPG devices, using convolutional denoising autoencoders in unsupervised transfer learning. Both the encoder and decoder employ a deep CNNs structure. The unique noise artifact problem that exists in AF detection is combated by utilising the multitask deep CNN architecture and transfer learning. DeepBeat improves the performance of AF detection from 0.54 to 0.96 in terms of F1 score in comparison with a single-task model. The unbalanced data problem that is common to biomedical applications can also be addressed. A novel framework [98] was proposed to identify noise segments based on modified frequency slice wavelet transform (MFSWT) representation and CNN classifier. MFSWT was used to transform each 1-D ECG segment into 2D time frequency (T-F) image; then, the image was fed into a 13-layer CNN model. The MFSWT method has the ability to better capture small changes in the frequency domain and provide quality image input for the CNN model. This model outperformed three other combination types from continuous wavelet transform (CWT) and ANN. In addition, an ECG diagnostic support system (EDSS) [84] was developed to detect ECG arrhythmia utilising deep 2D-CNN with images based on spectrograms. The advantages of using spectrogram images as input are that the visual examination such as identification of R-peak or P-peak is not needed, and is reliable. As the noise data are ignored, it is suitable for ECG devices that have various amplitudes and sampling rates. In [73], two novel DL models were developed for performing accurate, noise resistant and robust QRS complex detection. One model mainly consists of convolutional blocks and Squeeze-and-Excitation networks (SENet). The other model (CRNN) is composed of a hybrid convolutional and LSTMs network. However, the models can only predict QRS locations that are approximate to the R-peak, and cannot recognise ventricular flutter. A deep residual NN was developed for identifying patients with paroxysmal supraventricular tachycardia (PSVT) during normal sinus rhythm. The DL model demonstrates that the delta wave and QT interval are critical to identify the PSVT. Further studies are needed to evaluate continuous long-term ECG monitoring [78].


*(b.4) Hybrid approaches*


Hybrid methods provide a chance to make use of each individual approach. A stacked CNNs-BiLSTM-based approach [126] has been developed for investigating AF screening of ECG waveforms. CNNs enable automatic extraction of the features from data, and are naturally more adaptable to variations among different ECG waveforms. They use ECG or PPG signal as inputs while BiLSTM is used as a classifier. ECG morphologies such as the P-wave, QRS complex, T-wave, along with a signal recording, can be recognised; thus, the rhythm abnormality of AF is identified. Compared to unidirectional (Uni-LSTM) and BiLSTM architectures without CNNs, the performance of the proposed approach is improved. A deep model that combined CNNs and SNNs was developed to classify ECG beats for detecting ECG diseases [67]. Raw heartbeat data are used directly as input into a two-stage CNNs workflow that can save power, as most wearable devices are not able to carry out much complex pre-processing computing in daily real-time activities. First a CNN with two convolutional layers is used to detect if the beat is normal. If the beat is normal, the classification process is stopped; if not, a CNN with three convolutional layers is used to further classify the detailed beat types. The early stopping reduces the time and energy cost when dealing with normal ECG beats. Further, an energy efficient SNN model was proposed to further reduce energy consumption. In comparison to [66], not only VEB beat type but another main class of abnormal beat type are distinguished from the normal beat with similar accuracy. A DL model that is based on an ensemble NN was proposed to detect HFpEF using ECG signals of patients [56]. The results demonstrated its performance when predicting the development of HFpEF.

### 3.3. Types of Cardiovascular Disease and AI Methods

There are a wide range of different forms of heart disease. Most are diagnosed using a combination of multiple forms of data, including the patient’s reported symptoms, ECG, imaging data (from ultrasound, X-ray, MRI, etc.) and blood tests. The manual analysis of these data for the diagnosis of heart diseases is time-consuming. With the development of wearables and miniature devices that can accurately record time, frequency information such as RR interval, i.e., the time interval between two heartbeats, and the quick development of machine learning and artificial intelligence techniques, automated and timely detection of heart diseases has greatly improved. Figure 7 shows the efforts made to address various types of heart disease in recent years. In the following subsection, several main heart diseases are explored.


*(1) Cardiac arrhythmias and Atrial fibrillation (AF, A-fib)*


The classification and prediction of cardiac arrhythmias are some of the most wide-spread applications of ML and DL in cardiology. DL models have been developed to effectively suppress the false alarms in intensive care units (ICUs), while five separate alarm types can be recognised: the true alarms are identified using single and multimodal biosignals [97] to diagnose arrhythmias using ECG data streaming in a sequential manner [79]. The CNN-based DL classification method has been used to identify distinct types of arrhythmias using direct analysis of images based on spectrograms, without requiring visual examination by a clinician [84]. Another application is to use supervised learning to identify five types of cardiac arrhythmias with different risks [38]. It should be stressed that two research works have been published that used deep SNNs to identify arrhythmias in a power-saving way [66,67]. Further, numerous studies have been developed to address detection and predictive models for AF disorders due to their impact and clinical significance [20,21,25,26,32,33,36,41,42,43,45,49,50,53,57,59,62,73,75,82,85,86,87,88,91,92].

Atrial fibrillation (AF) is the commonest significant rhythm disorder, affecting 2–3% of adults. Although often AF causes no symptoms, it can cause disabling symptoms of breathlessness and an awareness of abnormal heart rhythm and increases the risk of stroke because of the mobilization of blood clots that form in the left atrium. A substantial proportion of AF is undiagnosed, making automatic AF detection based on ECG monitoring highly desirable. However, it is technically challenging to achieve accurate detection of AF episodes-based ECG signals. Figure 8 summaries research studies seeking to identify AF and the type of AI approach used.

Premature ventricular contraction (PVC), which is one of the most common ventricular arrhythmias, and paroxysmal supraventricular tachycardia (PSVT) have received attention in the medical field [78], and more research efforts are expected in the future.


*(2) Myocardial Infarction (MI)*


Myocardial infarction (MI), also called heart attack, is a major cause of death worldwide. Chest pain and breathlessness are the main symptoms. The risk factors of acute MI include family history, hypertension, smoking, obesity, cholesterol, diabetes, and low physical activity. In recent years, AI-based diagnostic tools for MI have been developed. A specialised deep Bidirectional Long Short-Term Memory (BiLSTM) architecture was presented to detect 12 heart rhythms based on 12-lead ECGs and acute ST-elevation myocardial infarction (STEMI) [60]. The proposed BiLSTM model achieved an accuracy of 0.987 for detecting STEMI, superior to both healthcare professionals and a commercial algorithm. An end-to-end deep MI framework was developed to classify MI and identify the time occurrence of MI as “Acute”, “Recent” and “Old” using raw ECG records [63]. A transfer learning technique that is used in existing computer vision networks was employed to extract features to minimize the computational overhead. RNNs are used to encode the time sequence information inherent in ECGs. Promising results were achieved (area under the ROC curves can reach 0.94).


*(3) Heart Failure (HF)*


Left ventricular systolic dysfunction (LVSD) can result in heart failure with reduced ejection fraction (HFrEF), with high mortality, reduced quality of life and longevity, and increased health care costs. A total of 3–6% of the adult population have LVSD. The ejection fraction (EF) measured by ultrasound is the most important indicator for predicting future complications. An AI-ECG CNNs system was developed to identify patients with ventricular dysfunction (EF ≤ 35%) and was evaluated on an external population [68,69,70]. This could be a useful screening tool to identify LVSD in asymptomatic individuals. Heart failure with preserved ejection fraction (HFpEF)is a major cause of mortality and health care expenditure in people over 65 years of age. It is estimated that its prevalence will continue to increase to 1.1–5.5% of the population [56]. ML- and DL-based AI medical diagnostic and predictive systems could improve outcomes [127]. Although it is challenging to develop a robust HF detection and risk prediction system, in recent years, several AI-based models have been developed for HF detection and prediction using diverse metrics generated from electronic health record (EHR) data that contain medical records, demographic, laboratory, and image data. For example, using traditional hierarchical clustering [46], similarity [52], ensemble-based [54] and deep learning approaches [56,71,74,89], comparable prediction results have been achieved. In [74], a deep learning method based on an ensemble neural network was developed, and validated in internal and external patients, respectively, with high performance demonstrated for HFpEF detection. Neural network-based variational autoencoders and hierarchical clustering have been applied to pooled individual patient data and the results demonstrate that fusing robust AI -based approaches provides potential clinical values to better identify responses to treatment for a fundamental therapy employed in patients with HF [46].


*(4) Hypertrophic Cardiomyopathy (HCM)*


About 30–40% of HCM patients experience left ventricular outflow tract obstruction (oHCM) at rest. There is a need for an effective tool to detect blood volume changes outside the clinical setting for real-time decision making. At the skin surface, photoplethysmography (PPG) that uses a non-invasive optical sensor can integrate with smart devices to detect blood volume changes. A multi-instance ML model was developed using PPG recordings and ECG collected from 19 HCM and 64 healthy volunteers [44]. Each recording was assigned an oHCM score based on qualified beats (instances). A set of 42 morphometric pulse wave features were extracted, and all beats in a recording were transformed into a single vector; then, SVM was used to fit the resulting vectors. The authors claim this method could act as a non-invasive and popular screening tool for obstructive HCM. While in [80], a CNN was developed using 12-lead ECG from 2448 HCM patients and 51,153 non-HCM control subjects. These results showed that the AI-based system could be used as a diagnostic tool after external validation and further refinement.


*(5) Coronary artery disease (CAD)*


Coronary artery disease (CAD) is also called ischemic heart disease or coronary heart disease. Plaque builds up in the coronary arteries and can cause angina and myocardial infarction. Thus, the early diagnosis of CAD is vital for the initiation of treatment and prevention of MI. An SVM-based ML model was developed to diagnose CAD and MI using data gathered from breath with an electronic nose [23]. A classification and regression tree (CART) model was developed in [22]; this DT-based algorithm is robust and achieved accurate and fast predictions, and the technique can be used to develop a system for decision-making regarding CAD diagnosis.

Several deep learning systems have been developed for the diagnosis of CAD. For example, myocardial perfusion imaging (MPI) has been used to predict automatically obstructive disease [72] and evaluated by comparison with total perfusion deficit (TPD). In [94], CNN and unique GaborCNN models were used to classify four classes (i.e., normal, CAD, myocardial infarction (MI) and congestive heart failure (CHF)). The imbalanced dataset is balanced using weight balancing. Compared to the CNN model, GaborCNN reduces computational complexity and is preferable. The authors claimed that the proposed model could be helpful for clinicians to diagnose CVDs using ECG signals. In addition, two other hybrid systems fusing CNN with LSTM have also been developed for CAD identification [65,95].


*(6) Valve disease*


Mitral regurgitation (MR), which causes blood flow from the left ventricle into the left atrium in a reverse direction, is the most common heart valve disorder, which gradually progresses and can lead to HF and death. A total of 1.7% of the general population in developed countries have significant MR, and approximately 10% of the population over age 70 are significantly affected [128,129]. A deep CNNs model to detect MR using 12-lead and single-lead ECGs obtained promising results [81].

## 4. Discussion

### 4.1. AI Algorithms and Models

There are a large amount of AI algorithms and models driven by data produced from wearable sensor devices. Classical machine learning and deep learning methods both play important roles in establishing diagnosis and prediction models using wearable sensor data.

Classical machine learning methods (Figure 4, Table 1) including supervised, unsupervised, ensemble and other rule-based or statistical approaches, have been deployed in ECG data processing and modelling (Table 1) in the investigated literature, relying on hand-engineered features. Domain knowledge is required for proper feature extraction and whilst many features are generated, feature selection methods are mostly needed. Among these, the RF and SVM are used in more studies (Table 1). Since these features vary significantly under different conditions and among individuals, deep learning has made great developments in recent years.

In deep learning processing, the use of ECG data is dominant (Figure 6, Table 2), including all types of lead ECGs. To obtain more information, the 12-lead ECG was mostly used. For the applied algorithms, LSTM-related modelling has been used in the diagnosis and prediction of CVD. Algorithms involving CNN form another focus of the application of AI in these practices, as they are usually applied to automatically extract time-invariant features, and the important segments of input signals are more emphasised. Long short-term memory units (LSTM) are applied to capture the temporal information that exists in the signal, and CNN and LSTM methods appear in the majority of the studies in the literature, showing their significance in wearable sensor data-based algorithms and models for the diagnosis and prediction of CVD. For different cardiovascular diseases, different types of algorithms can be applied in terms of their accuracy performance and efficiency (Table 1 and Table 2). Suitable algorithms and models for specific types of CVD need to be established, but, in general, the performance of deep learning algorithms is more appealing.

### 4.2. Functioning of Wearable Devices

*Monitoring*: wearable devices offer a way to continuously monitor health parameters such as heart rate and heart rhythm, etc., in a user-friendly, non-invasive way. The continuous monitoring of physiological parameters then offers a potential solution to more timely access to CVD-based healthcare [9,10,14,47,50,52,53,59,66].

*Diagnosis*: Physicians in their routine practices may benefit from the AI-based diagnostic tool to achieve accurate diagnosis and inform the accuracy of their analyses. Underpinned by AI models and the sensor data, timely diagnostics can be realised and can help to reduce mortality among various CVD populations [8,20,25,26,27,28,41,42,43,44,50,53,56,62,69,73,76,77,80,86,87,88,89,90,91,92,93,94,95,96,101,102,103,104,106,117,118,122].

*Management*: Wearables will assist in the transition of medical care and management from the clinical setting to the home [10]. Clinical benefits that may be obtained from the use of such devices include personalising AF management, refining stroke prevention strategies, and optimizing the patient–physician relationship, etc.

*Change*: Wearable devices are changing the future of therapeutic care and cardiovascular prevention as well as the way that clinicians perform their research. Diagnostic capabilities of remote ECG devices are rapidly developing. With advancements in artificial intelligence, the signal acquired from the devices will help realise the potential to accurately detect episodes of atrial fibrillation and may replace conventional diagnostic and long-term monitoring methods [130].

### 4.3. Challenges and Opportunities

Digital health technologies are being increasingly applied in the field of cardiology, bringing enormous opportunities for CVD care. However, there are several areas where wearable sensor technologies may have limitations [8,10,11,103,104,105,111,118].

*Improvement needed in technical aspects*: There has been a concern about device accuracy that requires more validation to provide the grounds for the further application of wearable devices. For example, although wearables may have been used in the initial detection of cardiac arrhythmias, it is significant to note that they have a restricted role for characterising such events. Technically, the ECG traces obtained from wearables do not directly correspond to those from the conventional Holter and 12-lead ECG monitoring systems. This should be investigated further. Researchers have pointed to the limitations of the PPG-based measurement of heart rate in heart-failure patients [10,131]. There is vulnerability in the PPG-based measurement of heart rates; thus, the measurement and monitoring need to pay particular attention to resting heart rate, which shows the best correlation with the ECG gold standard [10]. Accuracy differs among devices, and decreases significantly with increasing activity levels; also, during exercise, PPG from smartwatch devices tends to be more sensitive to motion artefacts than ECGs from chest straps [105].

*Management issues:* With the rapid development of sensor and computing technologies, wearables will become more versatile with more functions, and will become an integral part of devices employed in CVD practices. These devices need to be regulated via comprehensive evaluation systems and proper regulatory policies to ensure safety and efficacy [15,113,114,120].

In the absence of medical supervision, the quality of ECG traces may possibly be reduced. In the application of wearable sensor devices, large amounts of ECG traces will be generated and will require interpretation.

Another concern is in relation to patient privacy and cost [131]. Sensitive wearable data are subject to breaches. Aside from the multiple technical solutions, blockchain has been used to develop advanced interventions for the improvement of prevalent standards of medical data and personal health records, and in the management and processing of data [17]. This technique will be helpful to protect and safeguard personal information.

The CVD patient population place high demands on healthcare, with many being elderly and there being a high prevalence of people with low incomes. Both of these groups have the least access to digital technologies as they are unable to meet the cost of wearable devices, and literacy/IT literacy issues are common. These fundamental barriers need to be addressed to facilitate the deployment of wearables in these nonnegligible populations [15,113,114,120].

There are several regulatory and technical challenges to introducing such technologies to health care. Most commercially available wearable devices are marketed as “wellbeing” technologies, rather than medical devices. This means that the data they generate are not approved for medical decision making. There is little incentive for device companies to fund expensive clinical studies and regulatory approval if they can market them directly to users without such evidence.

Even if the use of such technologies can be shown to be beneficial in clinical studies, there is a substantial difference between proof-of-principle and implementation in complex healthcare IT infrastructures. The data and AI models will need to be made available to clinicians within their individual healthcare IT ecosystems, which differ widely across sectors and regions. The data and models will need to be updated regularly to be available at the point of decision-making, while still protecting privacy and data security. Although these challenges are significant, the investment required to overcome them will only be justified once proof of clinical benefit can be obtained.

*Application of the AI algorithms:* one of the challenges is the large amount of CVD related AI algorithms that have been developed using data from conventional 12-lead ECGs. Thus, the performance of the algorithms may vary when single-lead ECG data are fed. This instability could lead to distortions in applying AI algorithms to ECG data acquired from wearable sensors. Retaining and tuning of the algorithms may be needed to solve this issue of transportability of the algorithms. Other issues that need to be considered include how to separate actionable data from noise to improve data collecting accuracy; the interpretability of the AI algorithms may also raise challenges for physicians who are expected to diagnose the results of the data from wearables, as well as the challenges of integrating data from wearable devices with clinical data. Finally, the use of AI and wearable devices in the detection diagnosis and prediction of CVD will require wider adoption by clinicians and patients if they are to become more mainstream in the field [8,9,18,65,109,110,116].

## 5. Conclusions

Digital health-based healthcare solutions have been introduced in cardiology research and on-site practice with the rapid development of computing and telecommunication technologies. This enables dominant and recessive patients to take a more active role in their own care and has the potential to improve contemporary clinical care. AI algorithms and models for various type of cardiovascular-associated diseases are being developed, particularly ones suitable for wearable and portable devices.

The integration of wearable and AI technologies in the diagnostic practices of CVD is still in its infancy [11,105] and paves the way for future research. The potential of wearables to monitor cardiovascular events has been demonstrated in many of the current studies identified in this paper, and their deployment as a diagnostic tool for cardiovascular practice is hindered by the rare availability of realistic datasets and proper systematic and prospective evaluations. As demonstrated through the various studies, the deep learning algorithm provides a very good performance compared to the existing conversional analytical methods that merely use human visualization for biosignals, e.g., ECG or PPG signals [8], (Table 1 and Table 2).

AI- and ML-based approaches are superior to conventional statistical methods for predicting cardiovascular events, but the AI deep learning algorithms have been found to perform better that ML algorithms in many cases (Section 3.2, Table 1 and Table 2). Going forward, it is still necessary to evaluate the applicability and performance of algorithms in real-life cases. The future of cardiological medicine and early predictive treatment will depend on the advancement and ‘perfection’ of algorithms [131,132,133].

## Figures and Tables

**Figure 1 sensors-22-08002-f001:**
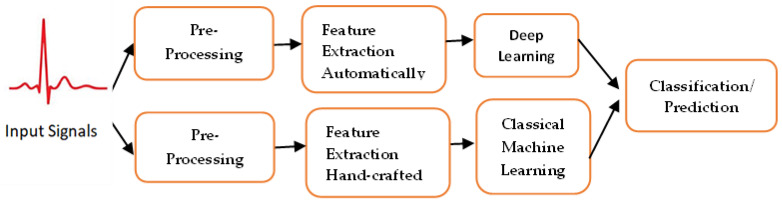
Deep learning-based methods that could extract features automatically, while classical machine learning needs human experts to extract features from the raw data, identify cardiac structural or functional abnormalities or make proactive predictions.

**Figure 2 sensors-22-08002-f002:**
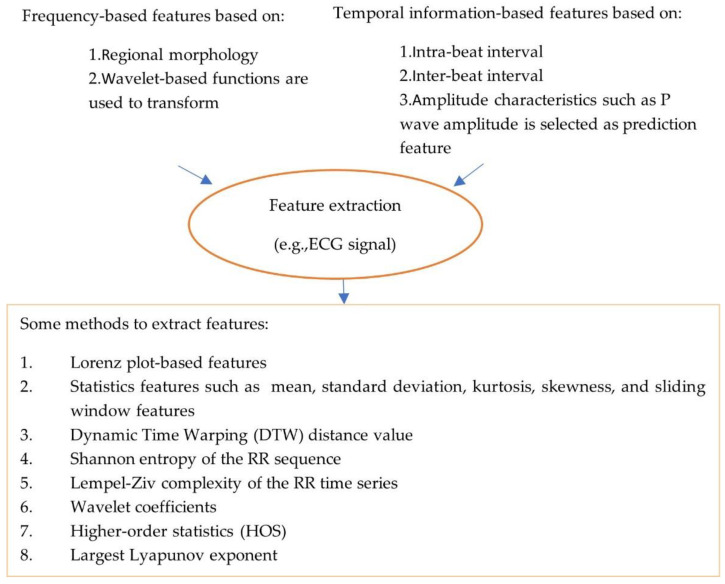
Illustration of types and methods of extracting features.

**Figure 3 sensors-22-08002-f003:**
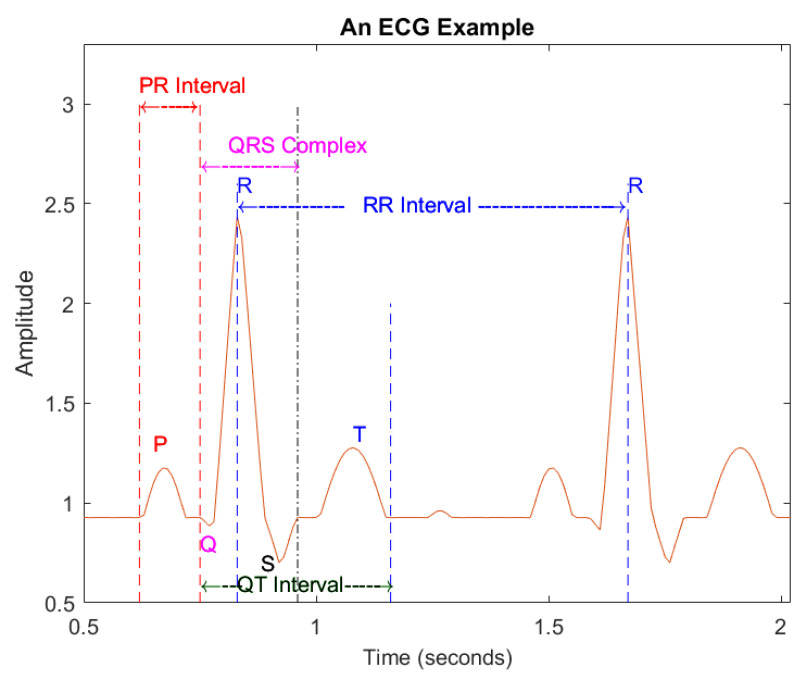
An example of ECG signals.

**Figure 4 sensors-22-08002-f004:**
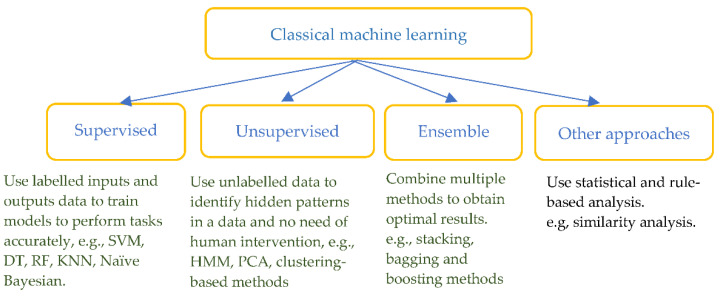
Classification of classical machine learning methods.

**Figure 5 sensors-22-08002-f005:**
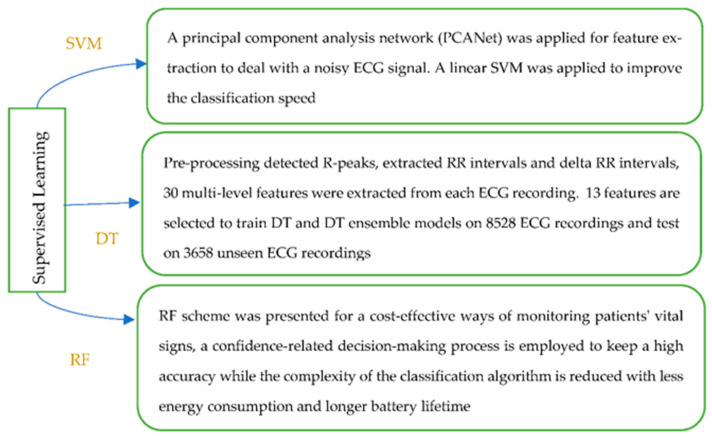
Supervised Machine Learning methods [20,28,38].

**Figure 6 sensors-22-08002-f006:**
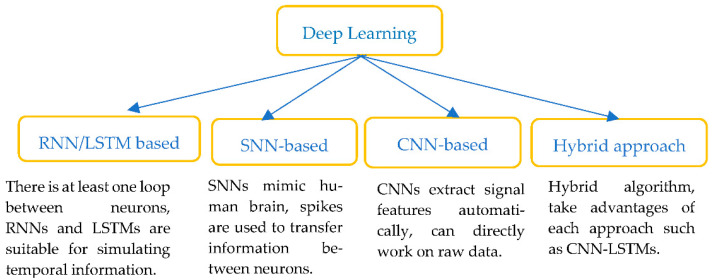
Classification of Deep learning methods.

**Figure 7 sensors-22-08002-f007:**
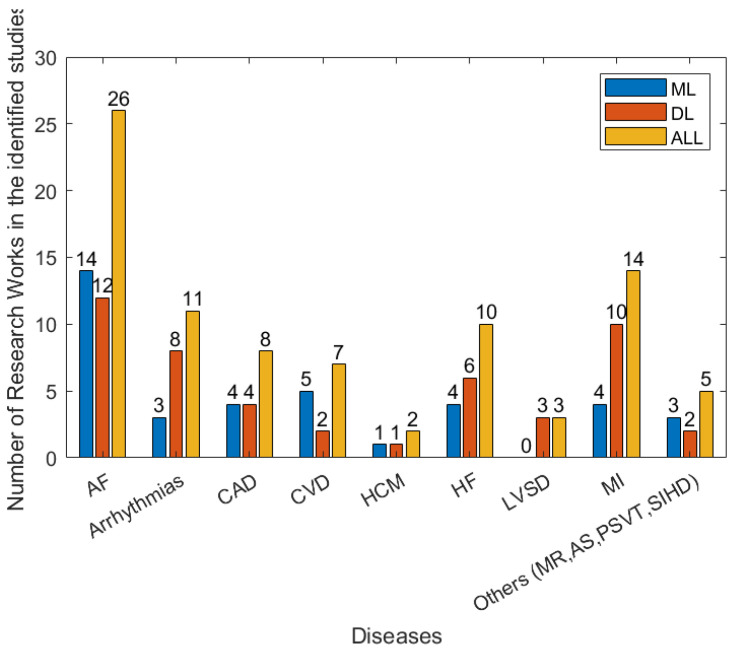
The numbers of types of disease in the 79 studies between January 2018 and April 2022 that are related to detection/prediction from the identified studies.

**Figure 8 sensors-22-08002-f008:**
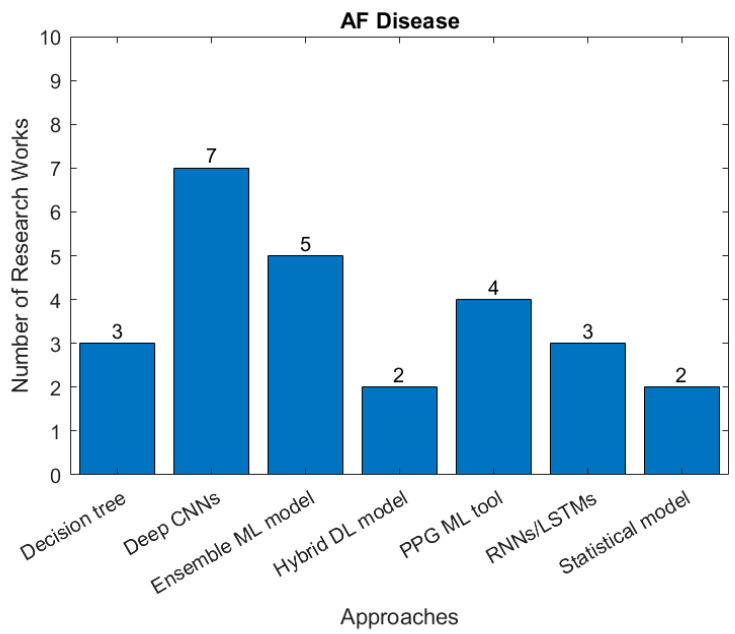
The number of studies applying each type of AI models for detection of AF (there are 14 papers related to ML, 12 related to DL in the identified papers between January 2018 and April 2022).

**Table 1 sensors-22-08002-t001:** Performance comparison of classical machine learning algorithms for wearable device datasets.

Author(s)/Database	Types of Diseases/Data	CML-Algorithms	Application	Evaluation
Shao et al. (2018) [20]/ 2017 PhysioNet CinC Challenge (CinC: Computing in Cardiology)	AF/ECG	DT, AdaBoosted DT ensemble	Classification (4 classes)	F1-score: 0.82
Fallet et al. (2019) [21]/ 17 patients (catheter ablation of cardiac arrhythmia)	AF and Ventricular arrhythmia/PPG, ECG, ACC-signals (ACC signals: three-axis accelerometer signals)	DT	Classification (2 classes)	ACC: 95.0% SPE: 92.8% SEN: 96.2%
Ghiasi et al. (2020) [22]/ Z-Alizadeh Sani CAD dataset: 303 patients	CAD/Databank: 55 independent parameters	DT-based CART (classification and regression tree)	Classification (2 classes)	ACC: 92.41%, TNR: 77.01%, TPR: 98.61%
Tozlu et al. (2021) [23]/ 33 MI patients, 22 CAD patients, 26 normal.	MI and CAD/ Electronic noses (19 gas sensors)	SVM	Classification (2 classes)	ACC: MI: 97.19%, CAD: 81.48%
Qureshi et al. (2020) [24]/ ~250 patients, Extracted CVD dataset	CVD/Physiological signals and clinical data	SVM and DT	Classification (2 categories)	ACC: 86.72%, SEN: 67.0%, SPE: 89.0%
Mei et al. (2018) [25]/ CinC 2017, (MIT-BIH AF) database	AF/ECG	SVM and Bagging trees	Classification (2 classes, 3 classes)	ACC: 92.0%-96.6% (Varies noise levels), 82.0% (3 classes)
Iftikhar et al. (2018) [26]/ 23 healthy people, 40 AF, 21 CAD, 21 MI patients	AF/SCG and GCG (seismo- and gyro- cardiogram-signals)	RF and SVM	Multiclass model (SR, AF, CAD, STEMI)	ACC: 75.24% F1: 74% (RF)
Sengupta et al. (2018) [27]/ 188 subjects	Abnormal Myocardial Relaxation (AMR)/spECG (spECG: Signal Processed Surface ECG)	RF/Monte Carlo cross-validation	Prediction	AUC: 91%, SEN: 80%, SPE: 84%
Sopic et al. (2018) [28]/ Physionet (PTB Diagnostic ECG database)	MI/ECG	RF	Classification and prediction	ACC: 83.26%, SEN: 87.95%, SPE: 78.82%
Meng et al. (2019) [29]/ Activity tracker data	SIHD/Tracker data	HMM	Output health status over time	AUC: 0.79
Akbulut & Akan (2018) [30]/ 30 participants	CVD/ECG	Decision Forest (DF), Logistic Regression (LR), NNs	Risk assessment	ACC: 96.0%
Dunn et al. (2021) [31]/ 54 integrative personal omics profiling (iPOP) participants	CVD/PPG, wVS HR, Electrodermal activity (EDA), physical activities	RF and Lasso models, canonical correlation analysis (CCA)	Prediction	wVS (wearable vital sigh) models outperform cVS (clinical vital sigh) models
Han et al. (2019) [32]/ 9530 controls, 306 cases	AF/AF burden signatures	Convolutional NN (CNN), RF and L1 regularized LR (LASSO)	Prediction of short-term stroke in 30-day window	AUC: RF: 0.662, Ensemble: 0.634
Hill et al. (2019) [33]/ CPRD (CPRD: UK Clinical Practice Research Datalink) 2,994,837 individuals (3.2% AF)	AF/ECG	Statistical/Models (NNs, LASSO, RF, SVM and Cox Regression)	Prediction	AUROC: 0.827 SEN: 75%
Jabeen et al. (2019) [34]/ UCI repository, 100 cardiac patients	CVD/Medical records	SVM, Naïve Bayes (NB), RF, Multilayer Perceptron (MLP)	Classification (8 classes)	ACC: 98% for Community-based heuristic approach
Kantoch E. (2018) [35]/ 5 participants, SPPB (SPPB: Short Physical Performance Battery task) test task	Sedentary Behavior (CVD risk)/Ambulatory and Daily activities	Linear Discriminant Analysis (LDA), DT, KNN, SVM, NB, Artificial NNs (ANNs)	Classification (6 activities)	ACC: 95.00% ± 2.11%
Kwan et al. (2021) [36]/ 50 participants	AF/PPG	XGBoost, RF, SVM and Gradient Boosting DT	Prediction	AF predicted 4 h in advance
Li, B. et al. (2019) [37]/ Hypertension patients, 3 datasets (stroke, HF, renal failure)	CVD/Medical records	Spark MLlib library (LR, SVM, NB)	A risk early warning model	LR(HF): AUC: 0.9269, ACC:0.8529, F1: 0.8456
Yang et al. (2018) [38]/ MIT-BIH arrhythmia Database	Arrhythmia/ECG	PCANet andand L-SVM, Back Propagation (BP)-NN, KNN	Identification (5 types)	ACC: 97.77% (skewed) 97.08% (noised)
Yang et al. (2020) [39]/ 20 AS patients, 20 health persons	AS/SCG and GCG	DT, RF and ANNs	Classification (2-classes, multi-classes)	ACC: (2/multi-classes): RF 97.43%/92.99%
Yang and Wei, (2020) [40]/ MIT-BIH AF database	Cardiac Arrhythmias/ECG	KNN, SVM and NNs	Classification (6 main types)	Best ACC: 97.70% (KNN)
Bumgarner et al. (2018) [41]/ 100 patients	AF/ECG	Kardia Band (KB) algorithm supported by Physician	Classification (2 classes)	SEN: 99%, SPE: 83%, K coefficient: 0.83
Dörr et al. (2019) [42]/ 672 participants	AF/PPG, iECG	Heartbeats PPG algorithm	Classification (2 classes)	ACC: 96.1%, SEN: 93.7%, SPE: 98.2%
Fan et al. (2019) [43]/ 112 participants	AF/Waveform recording from PPG	PRO AF PPG algorithm	Classification (2 classes)	Smart bands: ACC: 97.72%, SEN: 95.36%, SPE: 99.70%
Green et al. (2019) [44]/ 19 patients and 64 healthy volunteers	oHCM (with left ventricular outflow tract obstruction)/PPG	Multiple-instance ML model	Classification (2 classes)	SEN: 95%, SPE: 98%, C-statistic: 0.99
Guo et al. (2019) [45]/ 187,912 used smart devices	AF/PPG	Discrimination rule PPG algorithm	Prediction	Positive predictive value: 91.6% (95% CI: 91.5% to91.8%)
Karwath et al. (2021) [46]/ 18,637 patients (LVEF < 50)	HFrEF/ECG	Hierarchical clustering	Statistical analysis	Mean Jaccard score: 0·571 (SD 0·073; *p* < 0·0001)
Khan and Algarni, (2020) [47]/ UCI dataset https://www.kaggle.com/datasets, accessed on 15 April 2020.	Heart disease/LoMT (LoMT: Internet of Medical Things) Sensor data and medical records	MSSO-ANFIS	Prediction	ACC: 99.45%, PRE: 96.54%
Zeng et al. (2020) [48]/ PTB database:290 subjects, in which 148 patients with MI and 52 controls	MI/ECG	TQWT-VMD- Radial Basis Function (RBF)	Classification (2 classes)	ACC:97.98%
Perez et al. (2019) [49]/ 419,297 participants	AF/PPG, ECG patch	Irregular pulse notification algorithm	Identification	Positive predictive value: 84% (95% CI, 76 to 92)
Shao et al. (2020) [50]/ AFDB-2017, MIT-BIH AF (MITBIH-AFDB)	AF/ECG patch	CatBoost-based method	Classification (4 classes)	F1: 0.92
Spaccarotella et al. (2020) [51]/ 100 participants, 54 STEMI, 27 non-STEMI, 19 normal	Acute coronary syndromes/ECG	Cohen κ coefficient and Bland–Altman analysis	Earlier diagnosis	For STEMI: SEN: 93%, SPE: 95%
Stehlik et al. (2020) [52]/ 100 subjects	HF/PPG	Similarity-based	Prediction	SEN: 88%, SPE: 85%
Steinhubl et al. (2018) [53]/ 2659 participants	AF/ECG	Statistical analysis	Assessment	3.0% difference (immediate vs. delayed monitoring)
Samuel et al. (2020) [54]/ UCI repository Cleveland HF disease dataset: 303 patients	HF/Medical records	HNCL (HNCL: Hierarchical Neighborhood Component-based-Learning)/adaptive multi-layer networks (AMLN)	Prediction	ACC: 97.8%, SEN: 95.45%, SPE: 100%

**Table 2 sensors-22-08002-t002:** Deep-learning-based solutions for cardiovascular diagnosis.

Author(s)/Database	Types of Diseases/Data	DL-Algorithms	Application	Evaluation
Mohammad et al. (2022) [55]/ 139,288 patients	MI/Medical records	ANN	Prediction 1-year-all-cause-mortality after MI	AUC: 0.87, ACC: 77.1%, SPE: 76.3%, SEN: 84.6%
Kwon et al. (2021) [56]/ 32,671 ECGs of 20,169 patients	HFpEF/12-lead ECG	Ensemble NN	Detect HFpEF	AUC: [0.866 0.869]
Chocron et al. (2020) [57]/ 2891 patients, PhysioNet LTAF test database.	AF/ECG	ArNet (a deep RNNs)	Estimation of the AF burden (AFB)	Estimation error: EAF: 1.2% (0.1–6.7)%
Feng et al. (2019) [58]/ Physikalisch–Technische Bundesanstalt (PTB) database	MI/I-lead ECG	CNNs/RNN	Classification (2 classes)	ACC: 95.4%, SEN: 98.2%, SPE: 86.5%,
Saadatnejad et al. (2019) [59]/ MIT-BIH arrhythmia database	AF/ECG	Wavelet transform (WT)/ LSTM-RNNs	Classification (7 types)	ACC: 99.2%, SEN: 93.0%, SPE: 99.8%
Chang et al. (2021) [60]/ 35,981 patients	12 heart rhythms and STEMI/ECG	Deep BiLSTM	STEMI detection	ACC: 98.7%, AUC: 0.997, F1: 0.909
Dami and Yahaghizadeh (2021) [61]/ Four publicly available datasets	(Super)-ventricular ectopic beats/ECG	Deep Belief Network (DBN)/LSTM	Prediction in advance of a few weeks or months	ACC: DB1:88.74%, DB2: 93.24%, DB3-4: 80.41%
Faust et al. (2018) [62]/ MIT-BIH AF database	AF/ECG, HR signals	Deep LSTMs	Classification (2 classes)	ACC: 98.51%, SEN: 98.32%, SPE: 98.67%
Tadesse et al. (2021) [63]/ 17,000 patients Evaluation: PTB	MI/ECG	End-to-end deep learning (Dense-LSTM)	Classification (4 classes)	AUC: 94.0%
Lui et al. (2018) [64]/ Physionet PTB dataset, AF-Challenge 2017	MI/I-lead ECG	CNN/LSTM stacking	Classification (MI, healthy, other CVD, noisy)	SEN: 92.4%, SPE: 97.7%, PPV: 97.2%
Tan et al. (2018) [65]/ PhysioNet, 7 CAD and 40 normal subjects	CAD/ECG	CNN/LSTM	Classification (2 classes)	ACC: 99.85% PRE: 0.9985 F1: 0.9952
Amirshahi and Hashemi (2019) [66]/ MIT-BIH arrhythmia database	Arrhythmia Patterns/ECG	Deep SNNs	Classification (4 types)	ACC: 97.9%, SEN: 80.2%, SPE: 99.8%
Yan et al. (2021) [67]/ MIT-BIH arrhythmia database	Arrhythmia Patterns/ECG	CNNS/SNNs	Classification (2 to 4 classes)	ACC: 90.0%
Attia et al. (2019) [68]/ 44,959 subjects, tested on 52,870 patients	ALVD/Paired 12-lead ECG and transthoracic echocardiogram (TTE)	AI-ECG algorithm (CNNs-based)	Prediction (EF ≤ 35%)	AUC: 0.93, SEN: 86.3%, SPE: 85.7%, ACC: 85.7%
Attia et al. (2019) [69]/ 16,056 patients	LVSD/12-lead ECG	Deep-CNNs	Prediction (EF ≤ 35%)	AUC: 0.918, SEN: 82.5%, SPE: 86.8%, ACC: 86.5%
Attia et al. (2021) [70]/ 4277 subjects	LVSD/12-lead ECG	AI-ECG algorithm (CNNs-based)	Validation in an external population Prediction (EF ≤ 35%)	AUC: 0.82, SEN: 96.9%, SPE: 97.4%, ACC:97.0%
Bachtiger et al. (2022) [71]/ 1050 patients	HF (LVEF)/ 1-lead ECG	AI-ECG algorithm (CNNs-based)	Prediction	AUC:0.91 SEN: 91.9% SPE: 80.2%
Betancur et al. (2018) [72]/ 1638 patients	Obstructive CAD/MPI	DNN	Prediction	AUC: 0.80/0.76 SPE: 82.3/69.8 per patient/vessel
Cai and Hu (2020) [73]/ Four open-access ECG databases	AF/ECG	SENet, CRNN	Predict QRS locations	ACC: 99.0% F1: 99.0%
Cho et al. (2021) [74]/ Internal validation (IV): 2908 patients. External (EV): 4176 patients	HFrEF (EF < 40%)/12-lead ECG	CNNs	Detection	AUC: (IV/EV) 0.913/0.961 ACC: 77.5%/91.1%
Christopoulos et al. (2020) [75]/ 1936 participants	AF/ECG	CNNs/Statistical analysis	prediction	C statistics: 0.69 (95% CI, 0.66–0.72)
Han et al. (2021) [76]/ 97,742 patients	MI/ECG	Residual networks	Detection cardiac disorders	AUC: 12-lead: 0.880 1-lead: 0.768
Hannun et al. (2019) [77]/ 91,232 single-lead ECGs from 53,549 patients	Arrhythmias/ECG	End-to-end DNNs	Classification rhythm diagnoses	ROC: 0.97, F1:0.837 > 0.780 (Cardiologists)
Jo et al. (2021) [78]/12,955 patients with normal sinus rhythm	PSVT/ECG	Deep residual NNs	Identify patients with PSVT	ACC: 97%, SEN: 86.8%, SPE: 97.2%
Kiyasseh et al. (2021) [79]/ Four publicly available datasets	Cardiac Arrhythmias/ 1-lead ECG	CLOPS (Deep CNNs)	Diagnose in various continual learning (CL) scenarios	AUC: 0.796 (SD 0.013)
Ko et al. (2020) [80]/Train/Test HCM: 2,448/612 Control: 51,153/12,788	HCM/ 12-lead ECG	Deep CNNs	Classification (2 classes: HCM and control)	AUC: 0.96 SEN: 87% SPE: 90%
Kwon et al. (2020) [81]/ 38,393 patients	MR/ECG	Deep CNNs	Prediction	AUC: 0.816 (Internal) 0.877 (External)
Lai et al. (2020) [82]/ 55 consecutive AF patients	AF/ECG	CNNs	Classification (2 classes)	ACC: 93.1%, SEN: 93.1%, SPE: 93.4%
Li, G.Y. et al. (2019) [83]/412 Subjects Data1: medical records Data2: physiological parameters	CVD/ Medical records, disease’s metrics	Deep CNNs	Pulse-wave Classification (5 diseases)	ACC: Data1: 95.0% Data2: 88.0%
Panganiban et al. (2021) [84]/ 4 datasets from PhysioNet	Arrhythmia/ECG	Spectrograms Image 2D-CNNs	Classification (2, 5 classes)	ACC: 98.73% binary & 97.33% for quinary
Ribeiro et al. (2020) [85]/ 2,322,513 ECG records from 1,676,384 patients	AF/ECG	DNN Residual network	Classification (6-abnormality)	SPE: 1.000, SEN: 0.769, F1: 0.870
Tison et al. (2018) [86]/ 9750 (347) participants (with AF)	AF/HR (PPG)	DNN	Classification (2 classes)	C statistic: 0.97 (95% CI, 0.94–1.00; *p* < 0.001)
Torres-Soto and Ashley (2020) [87]/ Tr:(Synapse ID: syn21985690), Ambulatory dataset	AF/PPG	DeepBeat (transfer learning with autoencoder)	Classification (2 classes: AF and Sinus Rhythm)	SEN: 98.0%, SPE:99.0% F1 score: 0.93
Wasserlauf et al. (2019) [88]/ 7500 AliveCor users, 26 patients for validation	AF/ECG, HR, activity	DCNN	Classification (2 classes: AF and Sinus Rhythm)	SEN: 74.8%, SPE: 90.0%
Yao et al. (2020) [89]/ ~400 clinicians and 20,000 patients	Low ejection fraction (EF)/12-lead ECG from EHR	DL	Statistical analysis	To prospectively evaluate a novel AI screening tool
Zhao, Y. et al. (2020) [90]/ 667 STEMI ECGs, 7571 control ECGs	STEMI/ECG	Deep AI CNNs	Classification (2 classes)	AUC: 0.9954, SEN: 96.75%, SPE: 99.20%, ACC: 99.01%
Zhu et al. (2020) [91]/ 70,692 patients (aged ≥18 years)	AF/ECG	Deep CNN	Real-time analysis	F1 score: 0·887 AUC: 0·983, SEN: 0·867, SPE: 0·995
Chen et al. (2021) [92]/ MIT-BIH database	AF/ECG	multi-feature extraction/ CNNs	Classification (2 classes)	ACC: 98.92%, SPE: 97.04%, SEN: 97.19%
Cho et al. (2020) [93]/ 9536, 1301, 1768 ECGs of adult patients	MI/ECG (6, 12-lead)	DL/variational autoencoder (VAE)	Detection	AUROC: 0.880 (internal) 0.854 (external)
Jahmunah et al. (2021) [94]/ 92 healthy controls, 7 CAD, 148 MI and 15 CHF patients	CAD, MI, C-HF/ECG	CNN and GaborCNN	Classification (4 classes)	ACC: 98.5%
Lih et al. (2020) [95]/ 92 normal, 7 CAD, 148 MI, and 15 C-HF patients	CAD, MI, C-HF/ECG	Deep CNNs/LSTMs	Classification (4 classes)	ACC: 98.5%
Ma et al. (2020) [96]/ MIT-BIH AF (train), PhysioNet/CinC 2017, CPSC 2018 databases	AF/ECG	CNNs/SVM	Classification (2 classes)	(ACC for 30s ECG episodes) 98.48%/99.21%
Mousavi et al. (2020) [97]/ PhysioNet/CinC 2015	Arrhythmia/ABP, PPG, ECG	DL (CNNs+RNNs)	Suppress the false alarm, classification	SEN: 93.88%, SPE: 92.05%
Zhao, Z. et al. (2019) [98]/ Collected 1000 10s ECG segments	CVDs/ Smart ECG vest	MFSWT (MFSWT: Modified frequency slice wavelet transforms)/Deep-CNNs	Identify the noisy ECG segments (3 classes)	ACC: 86.3%, Kappa coefficient: [0.61 0.80]

## Data Availability

Not applicable.

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
