# Peer review of "Applying Artificial Intelligence to Wearable Sensor Data to Diagnose and Predict Cardiovascular Disease: A Review"

_sensors, 2022, doi:10.3390/s22208002_

Round 1

Reviewer 1 Report

The manuscript provides a systematic coverage of AI for wearable diagnostics applications. The content is interesting. A few suggestions are made to improve the quality of manuscript for publication

-Abstract should be improved. It is very short. It should highlight the significance of work and key findings from literature review.

-Introduction should be made more comprehensive to discuss AI applications from a broader perspective so readers can get idea of its versatility and wide scope. A few references are recommended in this context

https://doi.org/10.3390/joitmc8010045

https://doi.org/10.1016/j.fuel.2022.125409

-Review methodology should be described

-X-axis should be included in figure 8

-Figure caption should be standalone. Please mention years over which these research works are summarized in figure 8, and figure 9

-A discussion should be included on current status from commercialization perspective and challenges

Reviewer 2 Report

Let us express my interest in your work. Firstly, it has a quite big range of studies discussed. It is quite important for today life since the mode of life is hardly affecting the level of health of each individual. On the other hand, the recent advance in informational technologies and artificial intelligence may affect the quality of life because of automation, in particular in the area of healthcare, diagnostics and e-health. The possibility to gather enormous volumes of information in this particular area gives large opportunities to develop new techniques to study human health and improve level of life and knowledge to find early signs of disorders and diseases of human.

Remarks:

Most issues are dedicated to formatting  to be fixed.

1. The line 73 contains unnecessary “.” symbol at the beginning of the sentence.

2. The figures 1 and 2 have 2 issues for each item. The figures over exceed the limits (borders) of the text. These and all following figures should be placed within the left and right margin. Also, the figure names and description should be placed on the same page as the figure. The figures should be mentioned (declared) before they appear in text. I suggest removing figure 1 (it is not necessary in the context of your study) and move the figure name from the figure 2 to the following block of text. It is too.

3. Figures 3 and 4 should be stacked one above another. These are probably the most important figures in the whole paper and should be as clear (i.e. readable or visible) as possible.

4. The title of the subsection should be placed at least within 2 lines of text before the ending of the page. In case of section 3 (line 161), the title should be placed on the following page.

5. Figure 5 should be aligned with the text (left margin), also the figure name should be placed on the same page, as the image mentioned on the page. Also, the disambiguation of the methods should be made within the main text, not the image name.

6. The table 1 should have at least 1 block of text before its appearance. Hence, sentence on lines 201-203 should be placed just before the table 1.

7. The Table 1 is too wide.  The disambiguation of the methods should be made in text (at least in the footnote), not in the footer of the table. I strongly suggest removing or compacting the columns. For instance, column 2 and 3 can be joined, as well as 1 and 7 (authors of paper and data).

8. The figure 6 should be resized to fit the margins of the page.

9. The section b title (line 309) uses the different formatting as of previous one (section a).

10. The figure 7 should be aligned to the left border.

11. Table 2 have the same issue as the Table 1. The solution is the same as of Table 1.

12. Figures 8 and 9 should be stacked one above another.

13. The text within lines 475-488 seem to have different interval. Also, the sources on line 487-488 seem to be too long. I suggest splitting them in few parts to improve readability of the sentence.

14. The subsection 4, 5, 6 (lines 536, 552, 571 as follows) use different type of title formatting. Unless

it is done intentionally (more significant results/problems to be studied), the formatting should be same as of sections 1-3 above.

As far as there are a lot of issues addressed to formatting, I strongly encourage authors to double-check if there are any similar issues left within the text

15. The discussed paper is remarkably larger (2x) than most of the papers, so I highly encourage you to increase the length of the abstract. For my opinion, you can disclose the structure of the paper in the abstract, so after reading the abstract with detailed description of the study, it would be much easier to navigate through your paper because it would be helpful for your readers.

16. The tables 1 and 2 discuss the general usage of some methods. It would be a good idea to list the methods in their order of appearance on the research scheme. For instance: “Wavelet/LSTM”, where the upper item of the fraction is the detection method, the lower – the classification method.

Round 2

Reviewer 1 Report

The authors have adequately addressed the comments in the revised manuscript. The manuscript is recommended for publication.